# Essentials to Improve the Effectiveness of Healthy Aging Programming: Consideration of Social Determinants and Utilization of a Theoretical Model

**DOI:** 10.3390/ijerph20156491

**Published:** 2023-08-01

**Authors:** Katarina Friberg-Felsted, Michael Caserta

**Affiliations:** Gerontology Interdisciplinary Program, College of Nursing, University of Utah, Salt Lake City, UT 84112, USA

**Keywords:** older adults, social determinants, health behavior, behavior change, transtheoretical model

## Abstract

Older adult health, while partially determined by genetics, is mostly determined by behavioral and lifestyle choices. Researchers and interventionists develop and administer behavioral health interventions with older adults, and interventions are advertised in any number of settings, for example, by providers in healthcare settings and by activity directors in senior centers or assisted living facilities. However, previous studies and metanalyses indicate that many interventions targeting older adults are unsuccessful in recruitment or in retention. While providers and activity directors may assume older adults are unwilling to participate in behavioral change, in reality, low participation may be caused by erroneous design and administration. The objective of this manuscript is to recommend to creators and implementers of behavioral interventions for older adults that they focus on two critical considerations: the contextual perspective pertaining to healthy aging as well as an appropriately employed theoretical model that most effectively informs program design and implementation. In this commentary, we discuss how Prochaska and DiClemente’s Transtheoretical Model of Health Behavior Change may lead to more desirable outcomes as it considers that a person may be at any one of six stages of change, from pre-contemplation to maintenance. Currently, many behavioral interventions are targeted at individuals who are poised for action or in maintenance phases, ignoring those in earlier phases, resulting in limited overall success. Regarding viewing healthy aging in a contextual manner, determinants external to the individual may remain unnoted and unconsidered when designing or recruiting for a behavioral intervention. In conclusion, the integration of an intrapersonal health behavior model such as the Transtheoretical Model of Health Behavior Change, coupled with clearer considerations of the interplay of contextual factors operating in the lives of older adults, may allow for more effective design and implementation, as well as resulting in higher participation in behavioral interventions targeted toward older adults.

## 1. Introduction

Aging, even in the absence of chronic disease, is often associated with a variety of biological changes that can contribute to decreases in skeletal muscle mass, strength, and function. Such losses decrease physiologic resilience and increase vulnerability to catastrophic events. Yet these losses are not inevitable. Strategies for both prevention and treatment are necessary for the health and well-being of older adults [1] Evidence-based health promotion programs can help older adults manage a variety of chronic conditions and address behavioral risk factors [2,3]. A plethora of behavioral interventions are created for and implemented with older adults, and interventions are advertised in any number of settings, for example, by providers in healthcare settings and by activity directors in senior centers and assisted living facilities. Yet, many are unsuccessful.

### 1.1. Literature Review and Gap in the Literature

In a recent metanalysis, Forsat and colleagues [4] noted that older adults are underrepresented in research studies, and this has been the case for several decades [5]. This dearth negatively affects older adults, as exclusion from research continues ageist biases and leads to significantly worse health outcomes [6,7]. A plethora of studies illustrate that this occurs across a variety of conditions, ranging from Alzheimer’s disease to coronary syndromes [8,9,10,11,12]. The consequences are often severe, even fatal, as illustrated by a recent article published in *JAMA Surgery* [13].

For example, in a recent systematic review of the underrepresentation of older adults in cancer trials, the results showed that in the face of a reasonable amount of literature describing multifaceted barriers, only one intervention study aimed to increase enrollment of older adults in trials [14]. Florisson and colleagues also noted that older adults are not sufficiently included in clinical trials, and worse, there is no trend towards improvement of this problem [15]. While one recent trial suggested the use of pens as an incentive for older adult recruitment [16], we posit that changes must be made at a deeper level than this. Low recruitment and retention of older adults from research is a common, dangerous, and ageist practice, and these ageist biases are increasing over time [7]. Todd and colleagues [17] recommend exploring the reasons for, and developing measures to address, low recruitment of older adults in studies and trials.

### 1.2. Purpose

Much health promotion programming is still action oriented, not stage oriented. Utilizing stage-oriented programming that considers context (such as environmental and social forces) may more efficaciously support behavioral change. This paper posits that low participation of older adults in interventions may be caused by two flaws in the intervention design and application. Creators and implementers of ineffective behavioral interventions for older adults are not considering a contextual perspective pertaining to healthy aging nor are they appropriately employing theoretical models that most effectively inform program design and implementation. Identifying the behavior change techniques that are most effective in fostering positive change is necessary to develop the most effective and engaging interventions to improve participants’ health behavior [18,19,20,21]. The objectives of this manuscript are to (1) highlight the importance of considering that healthy aging is an interplay of multiple factors that operate throughout the life course, and (2) to illustrate the necessity of selecting a theory or model that lends itself to the idea that, due to these factors and others the interventionist may be aware of, communication and outreach must be sculpted for participants who reside along a spectrum of readiness. The purpose of this manuscript is to highlight crucial considerations to improve recruitment for behavioral interventions designed for older adults.

## 2. A Contextual View of Healthy Aging

Health is acknowledged as multidimensional [22]. For example, besides having behavioral factors and genetic dispositions, those at high risk for certain chronic conditions are often also low income, without health insurance, and have difficulty accessing healthcare. As such, a contextual view of healthy aging is not a conceptual model, but a shift in perspective. While genetics can set some predetermined parameters, for a more accurate perspective on healthy aging, components of contextual and social determinants must be taken into consideration.

Seven major components ought to be considered, and, perhaps more importantly, their interconnection needs to be examined [23,24,25,26]. Each of these factors can be inextricably interrelated to varying degrees.

## 3. Genetics

Genetics, the first, is mostly overestimated in its ability to affect an older person’s journey in optimal aging. Research has shown that only 25% of one’s aging journey is typically the result of genetics [27]; however, its role cannot be entirely discounted. While genetics can set parameters for risk, the extent to which genetic predispositions are expressed in health outcomes can be modified in varying degrees by lifestyle and/or environmental factors. Hence, if one is properly informed of genetic predispositions to adverse outcomes (e.g., family history), they can take preventive actions (e.g., vigilant screening, lifestyle changes, seeking protective environments) to minimize risk. These strategies are appropriate foci of healthy aging programming.

## 4. Lifestyle

Lifestyle arguably represents the most significant factor contributing to healthy aging, largely because it is the one most amenable to change. Research has shown that a vast variety of chronic conditions and co-morbidities correlate with the lifestyle of the older adult [28,29]. Behaviors related to nutrition, physical activity, alcohol or drug use/abuse, tobacco use, medication management, and how one copes with stress, among others, not only account for a large share of the influence on healthy aging outcomes but are also the most sensitive to intervention.

### 4.1. Social Influences

Consideration of factors in healthy aging that are external to the individual is also necessary. Social influences occur at the macro, mezzo, and micro levels. At the micro level these determine one’s value system as well as one’s attitudes about health, often as an outcome of the socialization process over the life course.

At a mezzo level are influences that are the result of an individual’s immediate social environment, such as family, friendships, and social networks. For example, marriage can be viewed through both a resource model lens and a stress model lens [30]. On one hand, it may provide financial security, social connectedness, and various forms of partner support; on the other, the stress of a marriage may contribute to poor health, abusive living situations, or other strain. Further, the partner may be highly supportive and reinforce a desired lifestyle change, or they could be reinforcing negative health behaviors. More recently, influences of social media can support or dissuade changes in health behavior [31]. Older adults have been identified as the “core audience of health information” on social media [32]. Shang and colleagues found that health-belief-related variables are associated with older adults’ likelihood of sharing health information on social media [33]. Perceived susceptibility is associated with higher rates of sharing, while perceived severity is associated with lower rates of sharing [33]. This access to health information has proven to facilitate health promotion. Other factors affecting an older adult on a mezzo level are social isolation and disconnectedness; both correlate with an increased risk of mental health, neurocognitive, and autoimmune disorders [34,35]. Events typical of later life, such as retirement, bereavement, and widowhood, affect perceived social support [36,37]. Social support is often correlated with health behavior.

At a macro level, societal influences affect the older adult in many ways, including the impact of health policy on an older adult’s life [38]. For instance, the introduction of Medicare’s coverage of prescription drugs had a positive effect on medication management, as older adults (particularly those on fixed incomes) are less likely to need to choose between obtaining needed medications (and taking them properly) and purchasing groceries, paying for housing or utilities. In addition to policy, cultural influences can come into play as they impact belief systems and health values, such as ethnic influences in diet and physical activity [39,40].

### 4.2. Environment

Environment may reinforce or inhibit the pursuit of healthy behavior. Sensitivity to and evaluation of an older adult’s environment would include several considerations. If an intervention recommends consumption of fresh vegetables and fruits, the older adult needs access to those foods to be able to participate. Not only would the physical environment need to contain a place where these can be purchased, that place would also need to be located in a walkable neighborhood. If the location was further than walking distance, the older adult would need available transportation that is accessible and safe. Environmental issues encompass more than available foods; as an example, this also affects the levels of pollution an older adult is exposed to. Another consideration of environment is whether their neighborhood is conducive to the behaviors they are seeking to change. If they want to exercise, the weather needs to be conducive, and/or a place to exercise would need to be accessible.

### 4.3. Economic Status

Similar to social and environmental factors, economic status is influential on health status and health behavior [41,42]. According to Pampel and colleagues [43], unhealthy behaviors in low socioeconomic status (SES) groups result from the differing social circumstances between high SES and low SES. Higher education is associated with greater health knowledge and greater engagement in health behaviors [44]. In some cases, it is noted that higher economic status correlates with overconsumption (of food, alcohol, and illicit drugs for example), which is a negative health behavior. Economic status often proportionately affects people’s access to resources for health [45].

### 4.4. Role of the Health Care System

Another contextual determinant of older adult health is the person’s relationship with the healthcare system, regarding whether a person does or does not have access to healthcare and the nature of that access [46]. The key here, besides access itself, is the extent to which one is an active participant in their own healthcare or a passive recipient of whatever the healthcare system provides them, often relinquishing control to the provider on health-related decisions. Active participation is based on a mutual partnership between an individual and his or her healthcare provider, where strong communication, taking part in decision making, and advocating for oneself are relevant factors in this relationship. There is a clear place for the healthcare system in healthy aging outcomes, and its role should be balanced with those self-care responsibilities individuals can assume for themselves within the environmental constraints in which they live [47].

### 4.5. Quality of Life

A final contextual determinant of health as we consider health promotion in an older adult population is the quality of life. Older adults are often able to maintain their functioning and independence, even in the face of chronic impairments. Regarding quality of life, it is most important to honor the older adult’s autonomy. They may make different health choices than seem subjectively better from another’s point of view [48]. 

The perspective of a multidimensional view of health allows for recognition of how each of these are interwoven. It is important to understand the interplay of factors associated with optimal aging and to facilitate this interplay.

It is common to note that while some older persons will engage in healthy behaviors, others will not. Some are ready, others are not ready, to act. Unfortunately, most behavior change interventions are designed for those who are prepared or ready for action. This readiness, and conversely, this unreadiness, explains why health promotion programs might not obtain the results they hope for. When designing and implementing behavioral change interventions, the use of a theoretical model applicable to individuals’ multiple stages of readiness may increase recruitment, decrease attrition, and improve a program’s success. Below, we present such a model.

## 5. How Utilization of a Theoretical Model Enhances Success

Prochaska and DiClemente’s Transtheoretical Model (TTM), Stages of Change, is ideal in that it addresses the stages of readiness of the individual [49]. This stage-matched approach represents a shift from an “action-oriented” to a “stage-oriented” paradigm. This approach requires that interventionists assess what stage the older adult is currently in and decide which processes to emphasize to facilitate them to the next stage. When dealing with a larger population, interventions need to speak to people at all levels, not only those who are ready for action.

The TTM is well validated in the literature, both in construct and concurrent validity [50,51,52,53,54,55]. Of relevance, the TTM has been utilized in numerous studies with older adult participants, including within a recent systematic review [56].

Notably, the TTM has been applied across a variety of disciplines, settings, and populations. Recent examples include occupational behavior, COVID-19 experiences, reproductive health, and health behavior interventions [57,58,59,60]. The TTM has been and continues to be used in international research, evidenced by recent translations of the model into Arabic and Malay [61,62].

The TTM outlines five stages of change, each building on the prior stage and each with its own timeline and features [63]. The model also recognizes that change is not linear, and that individuals can oscillate between stages at various times.

The first three stages emphasize the decision-making process; decisional balance (weighing pros and cons); and considering the costs versus benefits in cognitive, affective, and evaluative processes. Beginning in the third stage and continuing through the latter two, the emphasis is on strategies related to commitment, conditioning, environmental controls, and support, bolstering and supporting new behavior. With each stage, we provide ways the interventionist may best support the respective stage and the crucial question for the interventionist to consider.

### 5.1. Precontemplation

Precontemplation is the first stage in Prochaska and DiClemente’s TTM. In this stage, the individual cannot see a need for change and does not intend to change high-risk behaviors in the foreseeable future. People may remain in the precontemplation stage until 6 months before they start thinking about changing a behavior [63]. People are in the precontemplative stage for a variety of reasons. They may be underinformed or uninformed about the consequences of the risk behavior. At this stage, the benefits of the risky behavior far outweigh their negative effects.

Individuals in the precontemplative stage may be demoralized about their ability to change, often because the perceived barriers to change are too great to overcome. In this stage, individuals typically have low self-efficacy and are easily swayed, as their confidence in performing the health behavior and perceived ability to stop the risk behavior is low.

The health behavior change perspective In this stage is to introduce the need for change without expecting immediate action. It is also useful at this stage to provide information regarding to what extent they are at risk either due to genetic predispositions, current lifestyle, or environmental context; showing the benefits of the change; and providing ways to overcome barriers. Progress in this stage means getting the person to think about changing.

The core question in this stage therefore is, will they eventually give it some thought?

### 5.2. Contemplation

In the contemplative stage, the person is thinking about changing. While they recognize a need to change, they are not yet ready to act. Procrastination and ambivalence are powerful themes. There is an ambivalence. In this stage, the barriers versus the benefits of change seem fairly equitable. The individual considers the pros and cons of the behavior change; however, their sense of self efficacy is still low.

From a timeline perspective, the older adult may seriously consider changing in the next 6 months; however, this stage could last as long as two years. Intervention designers must focus on reducing the perceived barriers to change [64,65]. It is essential here to distinguish between perceived barriers that can be addressed through those processes which are characteristic of the early stages of the model versus structural constraints that are not mutable but need to be considered in a strategic way. It is also appropriate here to begin teaching how to go about beginning to change and still not have expectations for immediate action. Inspire them to begin taking the first step. One strategy is to suggest delaying engaging in a risky behavior instead of stopping altogether. For example, suggest a smoker wait a period of time before having their first cigarette as opposed to doing so immediately upon rising. Here, the core question is, will they take that first step?

### 5.3. Preparation

Those in the preparation stage are intending to act soon; in fact, they may have already taken some steps to action, but they are not regular or routine in their behavior. While they do not feel a firm commitment, at this stage the negatives of the high-risk behavior now outweigh the positives.

In the preparation stage, self-efficacy is increasing, but it remains a very unstable stage, one that could be described as dabbling. Here, the intervention needs to provide support to help move into the action phase, to prevent a falling back. The older adult has taken a few small steps, and they need to be presented with additional small, doable steps. Here, goal setting begins to become important.

The question in the preparation stage is, will they commit?

### 5.4. Action

As mentioned previously, many interventions fail to achieve intended outcomes if it is assumed the clients are ready to enter this stage, when in fact they may be in one of the previous stages. However, once they are in this stage, efforts must continue to help individuals move forward. For those actually in the action stage, serious steps have been taken toward change and visible changes in behavior are occurring. A specific criterion often defines the behavior (“I usually work out every other day”), and at the later edge of this stage, the individual has progressed to the point of established criteria (“I exercise five times per week, 30 minutes per day”). The action stage still has a sense of a precarious balance between self-efficacy and relapse to the original risk behavior. Here, the program must focus on maximizing self-efficacy in the individual and use reinforcement to protect the individual from relapse. In the action stage, the individual becomes increasingly self-aware of positive changes in themselves, which provides an internal motivation to continue. Here, the person is typically open to suggestions and advice to keep them moving. The goal of this stage is to see the behavior become a practice ingrained in their lifestyle. The question of the action stage becomes, will they adhere?

At the same time, it is acknowledged relapse can occur in which strategies associated with the earlier stages of the model can be employed. This a key advantage of programs based on the TTM, where strategies are available to help clients re-engage and “get back on track”, rather than merely lose them to attrition or failed adherence.

### 5.5. Maintenance

Maintenance is usually achieved after the individual has spent approximately 6 months in the action stage, since the criterion was reached and is continued. In the maintenance stage, the possibility of the individual returning to the risk behavior is significantly diminished. In the maintenance stage, the behavior becomes a practice. Self-efficacy is high. The question of the maintenance stage becomes, will they maintain?

The unique challenge of the maintenance stage is to keep the individual interested and motivated, keeping boredom at bay. Here, the key is to prevent the “now what” syndrome. Intervention must continue support, encouraging self-efficacy and providing reinforcement. In this stage, it is effective to help the individual to discover supportive environments conducive to this lifestyle, continually reinforcing outcomes of their action and noting the intrinsic rewards. A role shift may also occur in the maintenance stage. An individual who was the learner becomes the teacher. This is often seen with programs focused on sobriety, physical activity, and nutrition, but can be applicable in many contexts [66,67].

## 6. Conclusions

### 6.1. Applications

Effective health education [68] for an older adult population needs two often overlooked considerations: (1) how individuals naturally operate within a varied fabric of contextual determinants, and (2) a program based on theoretical foundations applicable to older adults, with diverse states of readiness to engage in behaviors amenable to healthy aging. We suggest that healthy aging programming be designed with these tenets in mind and regularly evaluated to improve the likelihood of success.

### 6.2. Limitations

This commentary addresses a specific situation arising within behavioral intervention research. It targets a specific population, older adults, and specific goals, improving recruitment and retention. This information may not be generalizable to interventions with other populations or with other goals.

### 6.3. Conclusions

It is noteworthy how complex encouraging a new health behavior is, or rather, how precise. Given that the scientific literature clearly illustrates that older adults are often overlooked in research, and that this oversight is severely detrimental, it is critical to create behavioral interventions that are attractive to the desired participant population. We have discussed two main ways to do so by focusing consideration on older adults’ contextual determinants as well as the individual older adult’s stage of readiness.

## Data Availability

Not applicable.

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
