# Peer review of "Essentials to Improve the Effectiveness of Healthy Aging Programming: Consideration of Social Determinants and Utilization of a Theoretical Model"

_ijerph, 2023, doi:10.3390/ijerph20156491_

Round 1
Reviewer 1 Report
The manuscript is a commentary about the transtheoretical model of health behavior change. The manuscript is written well and my comments are minor.
1. On page 3, Section 4.3, what is SES ?
2. On page 4, Section 4.5, paragraph 2: "Not only is it important..." should be "Not only it is important..."
3. Is Prochaska et al the only paper that describes the transtheoretical model ? Was this model validated ?
Author Response
Please see the PDF attachment

Reviewer 2 Report
I am grateful to be able to review the manuscript. It is an article of interest but needs considerable revision.
The abstract is not accurate and does not set clear objectives.
The study does not state the method used or the objectives and hypotheses accurately. The theoretical framework is obsolete and lacks structure and coherence. It is necessary to review in depth the contributions of the model to the audience. The authors need to state how they achieve the results.
The applications and limitations of the work should be improved.
Author Response
Please see the PDF attachment

Reviewer 3 Report
I am very honor to review this intresting issue. However, it seems to be missing too much information including literature review, research gaps, important conclusion which I have to reject it.
Concerning the abstract, you should present the important conclusion.
Concerning the introduction section, you should present the reason, motivation of researching this issue. In addition, you should present the research gaps, objects.
Overall, it does not look like academic paper.
Minor editing of English language required
Author Response
Please see PDF attachment

Round 2
Reviewer 2 Report
The manuscript has improved in the current version but it is still necessary to update the references used. It is also recommended that the authors clarify the innovative part of the study.
Reviewer 3 Report
Thank you very much for your detailed explanations. All question are well solved.
Minor editing of English language required
